# Molecular pathology of the R117H cystic fibrosis mutation is explained by loss of a hydrogen bond

Márton A Simon[1,2], László Csanády[1,2,3]*

[1]Department of Biochemistry, Semmelweis University, Budapest, Hungary; [2]HCEMM-SE Molecular Channelopathies Research Group, Budapest, Hungary; [3]MTA-SE Ion Channel Research Group, Semmelweis University, Budapest, Hungary

**Abstract** The phosphorylation-activated anion channel cystic fibrosis transmembrane conductance regulator (CFTR) is gated by an ATP hydrolysis cycle at its two cytosolic nucleotide-binding domains, and is essential for epithelial salt-water transport. A large number of CFTR mutations cause cystic fibrosis. Since recent breakthrough in targeted pharmacotherapy, CFTR mutants with impaired gating are candidates for stimulation by potentiator drugs. Thus, understanding the molecular pathology of individual mutations has become important. The relatively common R117H mutation affects an extracellular loop, but nevertheless causes a strong gating defect. Here, we identify a hydrogen bond between the side chain of arginine 117 and the backbone carbonyl group of glutamate 1124 in the cryo-electronmicroscopic structure of phosphorylated, ATP-bound CFTR. We address the functional relevance of that interaction for CFTR gating using macroscopic and microscopic inside-out patch-clamp recordings. Employing thermodynamic double-mutant cycles, we systematically track gating-state-dependent changes in the strength of the R117-E1124 interaction. We find that the H-bond is formed only in the open state, but neither in the short-lived 'flickery' nor in the long-lived 'interburst' closed state. Loss of this H-bond explains the strong gating phenotype of the R117H mutant, including robustly shortened burst durations and strongly reduced intraburst open probability. The findings may help targeted potentiator design.

*For correspondence:
csanady.laszlo@med.
semmelweis-univ.hu

Competing interest: The authors
declare that no competing
interests exist.

Reviewing Editor: Andrés Jara-
Oseguera, The University of
Texas at Austin, United States

## Editor's evaluation

Multiple inherited mutations in the epithelial CFTR anion-permeable channel cause cystic fibrosis through different molecular mechanisms, and some of these mechanisms can be specifically targeted by drugs to treat the disease. Drawing from available structural information and double-mutant cycle analysis applied to patch-clamp recordings, Simon and Csanády find that one of the most common CFTR disease-causing mutations, R117H, disrupts an interaction between the R117 side-chain and a main-chain carbonyl that selectively stabilizes the open state of the channel. These findings may open new paths of exploration for treating patients carrying this mutation, and provide important mechanistic constraints towards understanding the gating mechanism of CFTR channels.

## Introduction

Loss-of-function mutations of the cystic fibrosis transmembrane conductance regulator (CFTR) anion channel disrupt transepithelial salt-water transport in the lung, intestine, pancreatic duct, and sweat duct, and cause cystic fibrosis (CF), the most common inherited lethal disease among caucasians (*O'Sullivan and Freedman, 2009*). Based on their molecular consequences the several hundred identified CF mutations have been categorized into classes, such as those that impair synthesis of the

full-length CFTR polypeptide (Class I), processing and trafficking of the CFTR protein (Class II), channel gating (Class III), or anion permeation through the open pore (Class IV) (*De Boeck and Amaral, 2016*). Until recently symptomatic therapy remained the only available treatment option for CF patients, but this has been profoundly changed by a major breakthrough in the development of small-molecule drugs that target the CFTR protein itself. Treatments by 'corrector' drugs that improve processing and maturation of Class II mutant CFTR protein, and by the 'potentiator' compound Vx-770 (ivacaftor) which increases open probability ($P_o$) of Class III mutant channels, have led to significant symptomatic improvement for patients with Class II/III mutations (*Ramsey et al., 2011*; *Davies et al., 2018*), and have been approved by the FDA. Because the responsiveness to potentiators of different Class III mutants is greatly variable (*Van Goor et al., 2014*), understanding the molecular pathologies of such mutations bears strong clinical relevance.

CFTR is an ATP-binding cassette (ABC) protein which contains two transmembrane domains (TMD1, -2; *Figure 1A*, *gray*) and two cytosolic nucleotide-binding domains (NBD1, -2; *Figure 1A*, *blue* and *green*). These two ABC-typical TMD-NBD halves are linked by a cytosolic regulatory (R) domain (*Figure 1A*, *magenta*) which is unique to CFTR and contains multiple serines that must be phosphorylated by cAMP-dependent protein kinase (PKA) to allow channel activity (*Riordan et al., 1989*). CFTR pore opening/closure (gating) is linked to an ATP hydrolysis cycle at the NBDs and resembles the active transport cycle of ABC exporters (reviewed, e.g., in *Csanády et al., 2019*). In the presence of ATP single phosphorylated CFTR channels open into 'bursts': groups of openings separated by brief (~10 ms) 'flickery' closures and flanked by long (~1 s) 'interburst (IB)' closures. Upon ATP binding CFTR's NBDs associate (*Figure 1B–C*) into a tight head-to-tail dimer which is accompanied by a large rearrangement of its TMDs from an inward-facing (*Figure 1B*) to an outward-facing (*Figure 1C*) orientation. The stable IB closed state corresponds to an inward-facing TMD conformation with separated NBDs in which the channel gate, near the extracellular membrane surface, is closed (*Liu et al., 2017*). The bursting (B) state features a tight NBD dimer that occludes two ATP molecules in interfacial binding sites (sites 1 and 2), and an outward-facing TMD conformation in which a large lateral gap between TM helices 4 and 6 continues to connect the pore to the cytosol, bypassing the constricted ABC-transporter inner gate (*El Hiani and Linsdell, 2015*; *El Hiani et al., 2016*; *Zhang et al., 2018a*). The B state is a compound state that includes the fully open (O) and the flickery closed ($C_f$) states. For wild-type (WT) CFTR the majority of sojourns in the highly stable B state are terminated by hydrolysis of the ATP at site 2, which prompts NBD dimer dissociation and resets the channel into the inward-facing IB state (*Csanády et al., 2010*).

The cryo-electron microscopic (cryo-EM) studies have provided unprecedented structural insight, but have left several questions open. First, in the outward-facing CFTR structure (pdbid: 6msm) the external end of the pore is not quite as wide as would be required for passage of chloride ions. Nevertheless, at present that structure is the best available model for the O state (*Zhang et al., 2018a*). Second, the structure of the $C_f$ state is unknown although, based on its fast kinetics, the O↔$C_f$ transition is likely a localized movement. Third, the inward-facing structure (pdbid: 5uak) does not represent the closed state of an active channel gating in the presence of ATP, as it was obtained from unphosphorylated CFTR in the absence of ATP, and correspondingly contains density for the unphosphorylated R domain wedged in between the two unliganded NBDs and the cytosolic ends of the TMD helices. Nevertheless, functional studies suggest that the overall organization of the TMDs in the above two structures resembles those of the open and of the 'active' IB state, respectively (*Bai et al., 2011*; *Cui et al., 2014*; *Wang et al., 2014*). Thus, the former structure (6msm) will be referred to as 'outward-facing' or 'quasi-open', and the latter (5uak) as 'inward-facing' or 'closed' throughout this study.

Substitution of the conserved arginine at position 117, and in particular mutation R117H, is among the most common CF mutations (*Dean et al., 1990*; *Wilschanski et al., 1995*; https://cftr2.org/). In heterologous expression systems the R117H mutation does not impair surface expression (*Sheppard et al., 1993*; *Hämmerle et al., 2001*), but severely reduces whole-cell CFTR currents (*Sheppard et al., 1993*). Consistent with its location in the first extracellular loop (ECL1), replacement of the R117 side chain (*Figure 1A*, *blue dot*) causes a mild conductance defect suggesting that its positive charge is normally involved in recruiting extracellular chloride ions to the outer mouth of the pore (*Sheppard et al., 1993*; *Zhou et al., 2008*). However, the robust reduction of whole-cell currents is not explained by that small conductance defect, but rather by a strong gating defect (*Sheppard et al., 1993*) which

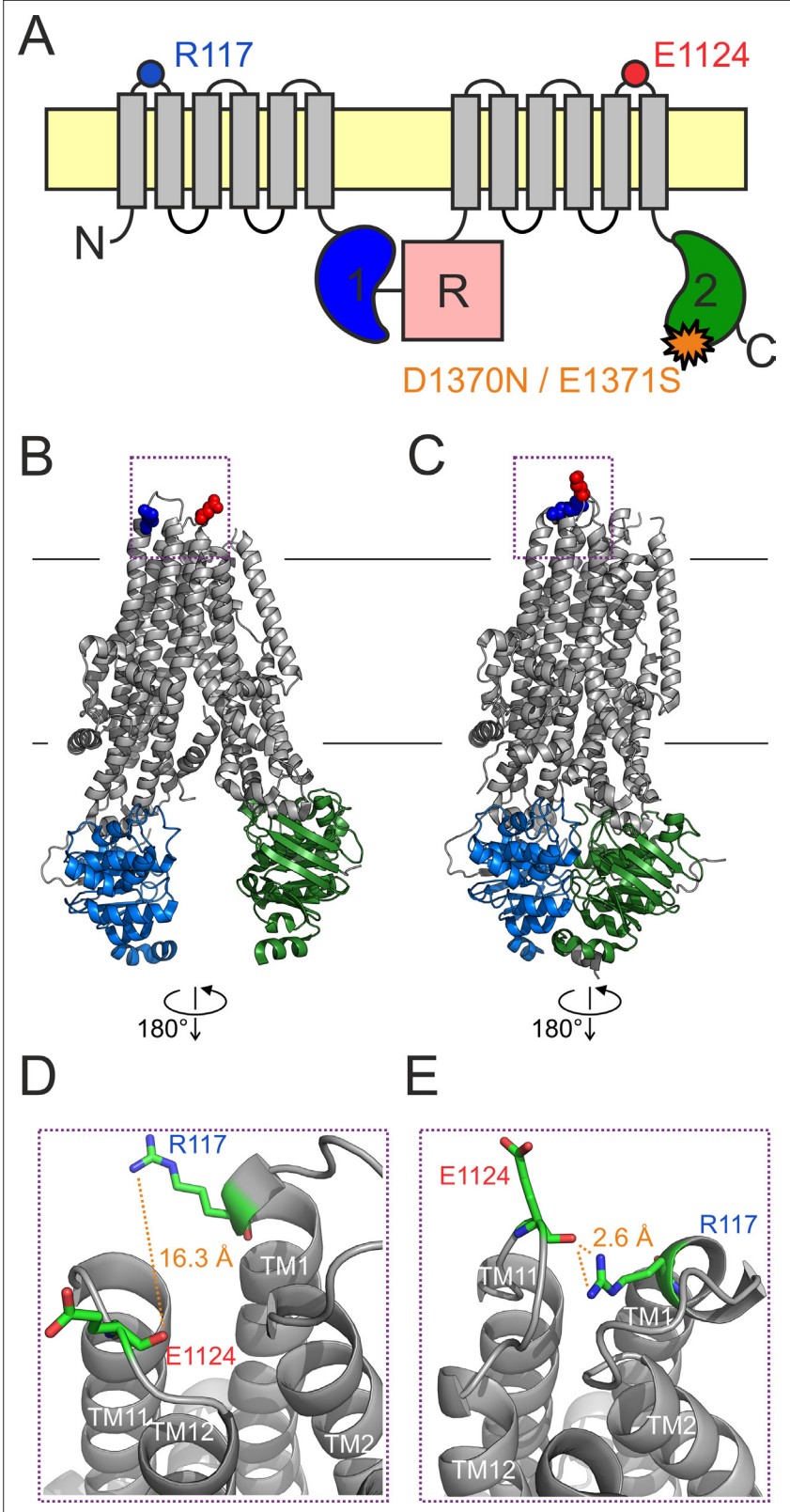

**Figure 1.** An H-bond between the R117 side chain and the E1124 backbone carbonyl group is apparent in the quasi-open CFTR structure. (A) Cartoon topology of CFTR and location of target positions (*blue* and *red dot*). TMDs, *gray*; NBD1, *blue*; NBD2, *green*; R domain, *magenta*; membrane, *yellow*. *Orange star* denotes catalytic site mutation in the NBD2 Walker B motif. (B–C) Target residues R117 (*blue*) and E1124 (*red*) shown in space fill on the

*Figure 1 continued on next page*

*Figure 1 continued*

cryo-EM structures of (**B**) inward-facing dephospho-CFTR (pdbid: 5uak) and (**C**) outward-facing phosphorylated ATP-bound CFTR (pdbid: 6msm). Domain color coding as in A, the R domain is not depicted. (D–E) Close-up views of the extracellular region highlighted in B–C by *dotted purple boxes*. Target residues are shown as *sticks*, target distances are illustrated by *orange dotted lines*.

The online version of this article includes the following figure supplement(s) for figure 1:

**Figure supplement 1.** Close-up view of the R117-E1124 interaction in the quasi-open human CFTR structure.

places the R117H mutation into Class III. The reduced $P_o$ of the mutant is primarily caused by a large reduction in the mean burst duration ($\tau_{burst}$) (*Sheppard et al., 1993*; *Yu et al., 2016*). In addition, the kinetics of intraburst gating is also altered in the mutant: shortened open times are paired with prolonged flickery closed times (*Yu et al., 2016*). Because R117H-CFTR currents are only modestly stimulated by Vx-770 (*Van Goor et al., 2014*; *Yu et al., 2016*), understanding the molecular pathology of this mutation will be important for developing better drugs optimized for personalized treatment.

How a mutation at the outer mouth of the pore may affect channel gating, thought to be controlled by NBD dimer formation/disruption, seemed puzzling at first. CFTR's cyclic gating mechanism precludes equilibrium thermodynamic analyses (*Csanády, 2009*), and makes it difficult to discern whether a mutational effect on burst durations reflects an alteration in the rate of ATP hydrolysis or in the stability of the prehydrolytic open state. Using a non-hydrolytic site 2 mutation which reduces CFTR gating in saturating ATP to reversible IB↔B transitions, Φ-value analysis (*Auerbach, 2007*) revealed that the pore opening (IB→B) step is a spreading conformational wave initiated at site 2 of the NBD dimer interface and propagated along the longitudinal protein axis toward the extracellular surface. In the high-energy transition state (T‡) site 2 has already adopted its B-state conformation (i.e., it is already tightly dimerized) whereas the extracellular portion of the TMDs, which includes the gate, is still in its IB-state (closed) conformation (*Sorum et al., 2015*). Thus, mutations at extracellular positions, like 117, are expected to similarly perturb the stabilities of the T‡ state and the IB closed state, and thus to selectively impact closing rate but not opening rate. That expectation was later confirmed for a series of perturbations at position 117: because Q, A, H, and C substitutions here all selectively accelerated closure (shortened $\tau_{burst}$), it was concluded that in WT CFTR the R117 side chain is involved in a stabilizing interaction that is formed only in the B state but neither in the IB state nor in the T‡ state for opening (*Sorum et al., 2017*). However, in the absence of high-resolution structures the interaction partner of the R117 side chain could not be identified.

Here, we exploit the recent cryo-EM structures of human CFTR obtained in inward- and outward-facing conformations (*Liu et al., 2017*; *Zhang et al., 2018a*), to identify the functionally important molecular interaction partner of the R117 side chain. We then test the functional role of this putative interaction using detailed single-channel kinetic analysis, and employ thermodynamic double-mutant cycles to rigorously quantitate how its strength changes between specific gating states. Our analysis provides a mechanistic explanation for the strong energetic role that the conserved arginine at position 117 plays in CFTR channel gating, as well as for the molecular pathology caused by its mutations. It also provides new structural inference for the organization of the $C_f$ state.

## Results

### Cryo-EM structures suggest a strong H-bond between the R117 side chain and the E1124 peptide carbonyl group in the open-channel state

In cryo-EM structures of both inward-facing (*Figure 1B*; *Liu et al., 2017*) and outward-facing (*Figure 1C*; *Zhang et al., 2018a*) CFTR, the R117 side chain is resolved (*Figure 1B–C*, *blue space fill*). In the quasi-open structure it approaches, and forms a strong H-bond with, the backbone carbonyl oxygen of residue E1124 (*Figure 1E*, *Figure 1—figure supplement 1*), located in ECL6 (*Figure 1A*, *red dot*). In the closed-pore structure it is pointing out toward the solvent, and is not in contact with any other residue of the protein (*Figure 1D*). Thus, based on the structures, the identified 117–1124 H-bond might represent an interaction that changes dynamically during the gating cycle. However, as these structures need to be interpreted with caution, we set out to study the functional relevance of this putative interaction for CFTR gating energetics. To study gating under equilibrium conditions, we

employed two different background mutations which disrupt ATP hydrolysis at site 2 (*Figure 1A*, *orange star*) and reduce CFTR gating in saturating ATP to a reversible IB↔B mechanism.

## Deletion of residue 1124 reproduces the R117 phenotype

The NBD2 Walker B glutamate (E1371) side chain forms the catalytic base in site 2, and its mutations disrupt ATP hydrolysis (*Zhang et al., 2018a*). Correspondingly, the E1371S mutation prolongs $\tau_{burst}$ by >100-fold (*Vergani et al., 2003*; *Csanády et al., 2013*), revealing slow reversibility of the IB→B gating step. That slow rate of the (non-hydrolytic) B→IB transition is conveniently measured in macroscopic inside-out patch recordings by activating pre-phosphorylated E1371S CFTR channels using a brief (~1 min) exposure to 5 mM ATP, and then observing the slow rate of macroscopic current decay upon sudden ATP removal (*Figure 2A*, *black trace*); the time constant of a fitted exponential reports $\tau_{burst}$ (*Figure 2B*, *black bar*), that is, the average life time of the B state. Introduction of the R117H mutation into this background robustly accelerated the current decay (*Figure 2A*, *blue trace*), yielding ~6-fold shortening of $\tau_{burst}$ (*Figure 2B*, *blue bar*), consistent with an earlier report using the E1371Q background (*Yu et al., 2016*).

If the phenotype caused by the R117H mutation is indeed due to loss of the R117-E1124 interaction, then a similar phenotype should be brought about by perturbations of position 1124. However, designing a suitable perturbation here is complicated by the fact that E1124 interacts through its peptide carbonyl oxygen. Notably, positioning of a backbone carbonyl group is unaffected by side chain substitutions, and its targeted elimination using nonsense suppression-based unnatural amino acid incorporation (*Pless and Ahern, 2013*) has not yet been developed. As expected, truncation of the E1124 side chain using the E1124A mutation did not affect closing rate (*Figure 2A*, *violet trace*; *Figure 2B*, *violet bar*), confirming that the E1124 side chain is not required for a normal burst duration. A slight shortening of $\tau_{burst}$ by the E1124G mutation suggested that increasing flexibility of ECL6 only marginally accelerates closing rate (*Figure 2A*, *green trace*; *Figure 2B*, *green bar*). We next examined whether insertion of a proline might distort ECL6 sufficiently to move the E1124 backbone carbonyl group out of reach of the R117 side chain. However, neither the E1124P (*Figure 2A*, *orange trace*; *Figure 2B*, *orange bar*) nor the E1126P (*Figure 2—figure supplement 1*) mutation caused a shortening of $\tau_{burst}$. Thus, shortening of the ECL6 loop using a single-residue deletion appeared to be the only viable strategy to increase the distance between the R117 side-chain guanidino group and the nearest ECL6 backbone carbonyl (in this case that of G1123). Indeed, in our E1371S background construct introduction of the E1124Δ mutation produced a phenotype very similar to that of R117H, accelerating the macroscopic current decay (*Figure 2A*, *red trace*), and shortening $\tau_{burst}$ by ~6-fold (*Figure 2B*, *red bar*).

## Energetic coupling between positions 117 and 1124 changes during non-hydrolytic closure

Functional effects of mutations of any individual amino acid residue in a protein provide little information regarding the role of a specific hypothesized residue-residue interaction, because in the WT protein the targeted residue might be involved in multiple interactions, all of which are altered by its mutations. To dissect and quantify the energetic effect of perturbing a specific residue-residue interaction, thermodynamic mutant cycle analysis must be employed (*Vergani et al., 2005*). The concept behind that approach is that the background construct, two single mutants that each perturb one or the other target residue, and the corresponding double-mutant form a thermodynamic cycle. If the two target residues do not interact, or if their interaction remains static (does not change between various gating states), then the energetic effects of mutating either target residue will be additive in the double mutant. Conversely, if the strength of the interaction between the target residues changes between gating states $S_1$ and $S_2$, then the effect on the relative stability of state $S_2$ caused by mutating one target residue ($\Delta\Delta G^0_{S2\text{-}S1}$) will depend on the nature of the residue present at the other target position – thus, the energetic effects of the two target-site mutations will not be additive in the double mutant. The change in strength of the target interaction during step $S_1 \rightarrow S_2$ ($\Delta\Delta G_{int} (S_1 \rightarrow S_2)$) is obtained as the difference between $\Delta\Delta G^0_{S2\text{-}S1}$ values along two parallel sides of the mutant cycle (*Figure 2E*).

Introducing the R117H mutation into an E1124Δ background caused only marginal further acceleration of E1371S non-hydrolytic closing rate (*Figure 2C*, *green trace*), shortening $\tau_{burst}$ ~1.47-fold

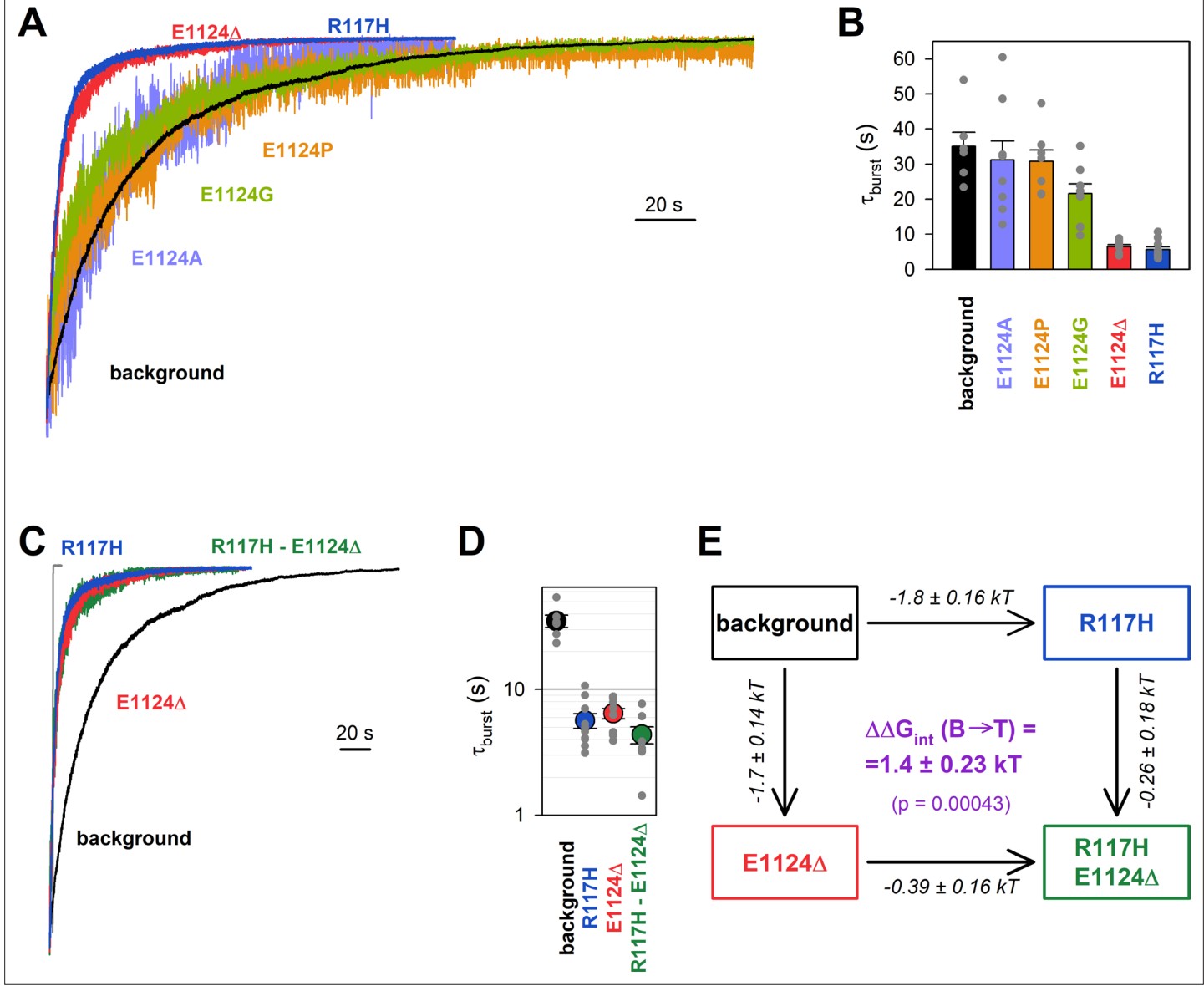

**Figure 2.** Energetic coupling between positions 117 and 1124 changes during non-hydrolytic closure. (A, C) Macroscopic current relaxations following ATP removal for indicated CFTR mutants (*color coded*) harboring the E1371S background mutation. Inside-out patch currents were activated by exposure of pre-phosphorylated channels to 5 mM ATP. Current amplitudes are shown rescaled by their steady-state values. The *gray trace* in C illustrates the speed of solution exchange, estimated as the time constant of deactivation (~100 ms) upon $Ca^{2+}$ removal of the endogenous $Ca^{2+}$-activated chloride current, evoked by brief exposure to $Ca^{2+}$. Membrane potential ($V_m$) was –20 mV. (B, D) Relaxation time constants of the currents in A and C, respectively, obtained by fits to single exponentials. Data are shown as mean ± SEM from 6 to 10 experiments. Note logarithmic ordinate in D. (E) Thermodynamic mutant cycle showing mutation-induced changes in the height of the free enthalpy barrier for the B→IB transition ($\Delta\Delta G^0_{T^{++}-B}$, *numbers on arrows; k*, Boltzmann's constant; *T*, absolute temperature). Each corner is represented by the mutations introduced into positions 117 and 1124 of E1371S-CFTR. $\Delta\Delta G_{int}\left(B \to T^{++}\right)$(*purple number*) is obtained as the difference between $\Delta\Delta G^0_{T^{++}-B}$ values along two parallel sides of the cycle.

The online version of this article includes the following figure supplement(s) for figure 2:

**Figure supplement 1.** The E1126P mutation slows channel closure.

**Figure supplement 2.** Sequence alignment of the ECL1 and ECL6 loops for CFTR orthologs.

(*Figure 2D*, *green* vs. *red symbol*). In energetic terms, the ~6-fold shortening of $\tau_{burst}$ brought about by the R117H mutation alone (*Figure 2D*, *blue* vs. *black symbol*) signals an ~1.8 kT decrease in the height of the free enthalpy barrier for the B→IB transition ($\Delta\Delta G^0_{T^{++}\text{-}B}$; *Figure 2E*, number on *top horizontal arrow*). In contrast, the ~1.47-fold acceleration caused by the same mutation in the E1124Δ background reports a decrease in the same barrier height by only ~0.39 kT (*Figure 2E*, number on *bottom horizontal arrow*). The difference between those two numbers quantifies the change in the strength of the R117-E1124 interaction in an E1371S channel while it progresses from the B state to the T$^{\ddagger}$ state, on its way toward the IB closed state (*Figure 2E*, *purple number*). That value of $\Delta\Delta G_{int}\left(B \rightarrow T^{++}\right)$ = 1.4 ± 0.23 kT is significantly different from zero (p = 0.00043), and its positive signature reports that a stabilizing interaction present in the B state is lost in the T$^{\ddagger}$ state. (The theoretical alternative of a lack of interaction in the B state but a destabilizing interaction in the T$^{\ddagger}$ state is inconsistent with structural evidence.)

## The stabilizing interaction between the target positions is present only in the bursting state

To address how the 117–1124 interaction changes between the IB state and the T$^{\ddagger}$ state, mutational effects on opening rate need to be quantified in steady-state single-channel recordings. Because gating of the E1371S background construct is prohibitively slow for steady-state single-channel analysis, we employed a different non-hydrolytic background mutation, D1370N, which perturbs the Walker B aspartate side chain that coordinates catalytic Mg$^{2+}$ in site 2 (*Hung et al., 1998*; *Rai et al., 2006*; *Gunderson and Kopito, 1995*). This mutation disrupts ATP hydrolysis but results in shorter burst durations, allowing steady-state kinetic analysis of the IB↔B gating process (*Csanády et al., 2010*; *Sorum et al., 2015*). The R117H, E1124Δ, and R117H/E1124Δ target-site mutations were introduced into this background, and gating of single prephosphorylated channels was studied in the presence of 5 mM ATP (*Figure 3A*), a saturating concentration for all four constructs (*Figure 3—figure supplement 1*). In such steady-state recordings flickery and IB closures can be readily discriminated by dwell-time analysis (Materials and methods; *Figure 3—figure supplement 2*), allowing calculation of not only $\tau_{burst}$, but also of mean IB durations ($\tau_{interburst}$) (*Table 1*).

Mutational effects on closing rate resembled those observed in the E1371S background: both the R117H and the E1124Δ mutation robustly shortened $\tau_{burst}$ (by ~30-fold and ~20-fold, respectively, *Figure 3B*, *blue* and *red* symbols, respectively, vs. *black symbol*), whereas little further shortening was observed in the double mutant (*Figure 3B*, *green symbol*). Thus, whereas the ~30-fold shortening of $\tau_{burst}$ by the R117H mutation alone (*Figure 3B*, *blue* vs. *black symbol*) reports a decrease in $\Delta\Delta G^0_{T^{++}\text{-}B}$ by ~3.4 kT (*Figure 3C*, *top horizontal arrow*), the ~1.76-fold shortening of $\tau_{burst}$ by the same mutation in the E1124Δ background (*Figure 3B*, *green* vs. *red symbol*) reports a decrease in the height of the same barrier by only ~0.57 kT (*Figure 3C*, *bottom horizontal arrow*). The disparity between those two values, $\Delta\Delta G_{int}\left(B \rightarrow T^{++}\right)$ = 2.8 ± 0.23 kT (*Figure 3C*, *purple number*), is significantly different from zero (p = 0.000034) confirming disruption of a strong stabilizing interaction between the target positions during the B→T$^{\ddagger}$ transition.

In contrast, channel opening rates were little affected by the target-site mutations, with $\tau_{interburst}$ values remaining within 2-fold for all four constructs (*Figure 3D*, *Table 1*). Correspondingly, the calculated change in the strength of the 117–1124 interaction during step IB→T$^{\ddagger}$, $\Delta\Delta G_{int}\left(IB \rightarrow T^{++}\right)$ (*Figure 3E*, *purple number*), was not significantly different from zero. Of note, correct estimation of $\tau_{interburst}$ strongly depends on correct estimation of the number of channels in the patch, which becomes increasingly difficult when the P$_o$ is very low. Thus, for our target-site mutants and the double mutant the presence of more than one channel in the patch was excluded by strongly boosting channel P$_o$ at the end of each recording using $N^6$-(2-phenylethyl)-dATP (P-dATP) and/or the potentiator drug Vx-770 (*Figure 3—figure supplement 3*, also see Materials and methods). Nevertheless, compared to $\tau_{burst}$, estimates of $\tau_{interburst}$ necessarily retain some degree of uncertainty, as the number of active channels can never be determined with 100% confidence (cf., *Yu et al., 2016*).

Taken together, these steady-state data reveal that in D1370N channels a strong stabilizing interaction between positions 117 and 1124 is formed in state B, but is absent in states T$^{\ddagger}$ and IB (*Figure 3F*, *purple* bond energy profile; *Figure 3F bottom*, cartoon).

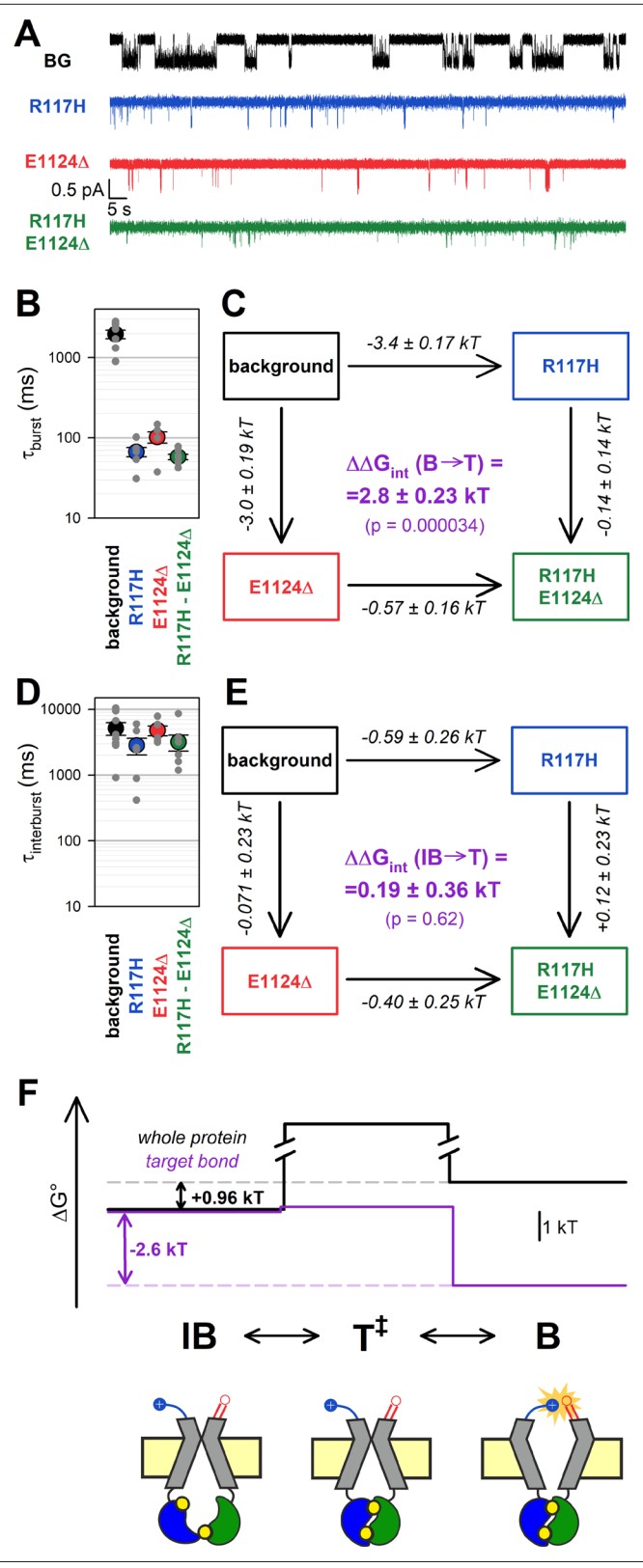

**Figure 3.** The H-bond between the target positions is formed only in the bursting state. (A) Single-channel currents in 5 mM ATP of indicated CFTR mutants (*color coded*) harboring the D1370N background (BG) mutation. Channels were pre-phosphorylated by ~1 min exposure to 300 nM PKA +5 mM ATP. $V_m$ was –80 mV. (B, D) Mean burst (**B**) and interburst (IB) (**D**) durations obtained by steady-state dwell-time analysis (see Materials and methods).

*Figure 3 continued on next page*

*Figure 3 continued*

Data are shown as mean ± SEM from five to eight patches; note logarithmic ordinates. (C, E) Thermodynamic mutant cycles showing mutation-induced changes in the height of the free enthalpy barrier for the (**C**) B→IB and (**E**) IB→B transition ($\Delta\Delta G^0_{T^{\ddagger}\text{-}B}$ and $\Delta\Delta G^0_{T^{\ddagger}\text{-}IB}$, respectively, *numbers on arrows; k*, Boltzmann's constant; *T*, absolute temperature). Each corner is represented by the mutations introduced into positions 117 and 1124 of D1370N-CFTR. $\Delta\Delta G_{int}$ (*purple number*) is obtained as the difference between $\Delta\Delta G^0$ values along two parallel sides of the cycle. (F) (*Top*) Free enthalpy profile of the entire channel protein (*black curve*) and of the 117–1124 interaction (*purple curve*) along the states of the IB↔B gating process (see Materials and methods). (*Bottom*) Cartoon representation of channel conformations in the IB, T$^{\ddagger}$, and B states. *Color coding* as in *Figure 1A*; ATP, *yellow circles*. *Blue* and *red* moieties represent the R117 side chain and the E1124 backbone carbonyl group, respectively. *Yellow star* in state B represents formation of the H-bond.

The online version of this article includes the following figure supplement(s) for figure 3:

**Figure supplement 1.** All tested constructs in the D1370N background are saturated by 5 mM ATP.

**Figure supplement 2.** Steady-state dwell-time distributions of the target-site mutants in the D1370N background.

**Figure supplement 3.** Stimulation by P-dATP or P-dATP+ Vx-770 facilitates counting channels for low-$P_o$ mutants.

## The H-bond between the target positions is broken in the flickery closed state

The bursting state comprises the open (O) and flickery closed ($C_f$) states. If the strength of an interaction changes between states O and $C_f$ then perturbing that interaction should affect intraburst $P_o$ ($P_{O|B}$), that is, the fraction of time the pore is open within a burst. To address whether the R117-E1124 H-bond is disrupted in the flickery closed state, we studied additivity of effects of the R117H and E1124Δ mutations on the intraburst closed-open equilibrium constant $K_{eq|B}$, obtained as $K_{eq|B} = P_{O|B}/(1 - P_{O|B})$. Effects on intraburst gating kinetics are most conveniently studied in the E1371S background, by briefly applying ATP to activate the channels and then focusing on the last bursting channel following ATP removal (*Figure 4A*, *Csanády et al., 2010*; *Yu et al., 2016*). Because in the absence of bath ATP reopening from the IB state is no longer possible, in such 'last-channel' segments of record all closures (except for the final) are necessarily flickery closures, and the long life time of the B state in E1371S allows sampling of a large number of O↔$C_f$ transitions.

As reported previously (*Yu et al., 2016*), the R117H mutation caused a strong reduction in $P_{O|B}$, due to an ~20-fold shortening of mean open times ($\tau_{open}$) and a 5-fold prolongation of mean flickery closed times ($\tau_{flicker}$) (*Figure 4A*, *insets*, *blue* vs. *black trace*; *Table 2*). Interestingly, this aspect of the R117H phenotype was also reproduced by the E1124Δ mutant, which demonstrated a similarly lowered $P_{O|B}$ (*Figure 4A*, *red trace*, *Table 2*). Although $P_{O|B}$ was even further reduced for R117H-E1124Δ (*Figure 4A*, *green trace*, *Table 2*), the effects of the two single mutations on $K_{eq|B}$ were not additive in the double mutant. Whereas the ~90-fold reduction in $K_{eq|B}$ caused by the R117H mutation alone (*Figure 4B*, *blue* vs. *black symbol*) reported an increase in the $C_f$-O free enthalpy difference ($\Delta\Delta G^0_{O\text{-}Cf}$) by ~4.5 kT (*Figure 4C*, *top vertical arrow*), the same mutation reduced $K_{eq|B}$ only ~10-fold (*Figure 4B*, *green* vs. *red symbol*), amounting to $\Delta\Delta G^0_{O\text{-}Cf} = $ ~ 2.3 kT (*Figure 4C*, *bottom vertical arrow*), when introduced into an E1124Δ background. The obtained interaction energy, $\Delta\Delta G_{int}$ ($C_f \rightarrow O$) = 2.2 ± 0.34 kT (*Figure 4C*, *purple number*), is significantly different from zero (p = 0.0017) indicating that, even

**Table 1.** Model-independent descriptive gating parameters for the indicated mutant CFTR constructs in the D1370N background.

Gating parameters were obtained from steady-state recordings as described in Materials and methods. Data are displayed as mean ± SEM from five to eight patches.

| | $\tau_{burst}$ (ms) | $\tau_{interburst}$ (ms) | $\tau_{flicker}$ (ms) | $n_{flicker}$ | $P_o$ |
|---|---|---|---|---|---|
| Background (D1370N) | 2000 ± 250 | 5100 ± 1100 | 10 ± 1.6 | 4.8 ± 0.82 | 0.35 ± 0.056 |
| R117H | 67 ± 8.8 | 2800 ± 800 | 33 ± 5.0 | 1.2 ± 0.12 | 0.017 ± 0.0055 |
| E1124Δ | 100 ± 16 | 4800 ± 810 | 23 ± 4.0 | 2.1 ± 0.41 | 0.013 ± 0.0033 |
| R117H E1124Δ | 58 ± 4.4 | 3200 ± 880 | 50 ± 7.9 | 1.1 ± 0.099 | 0.0054 ± 0.0011 |

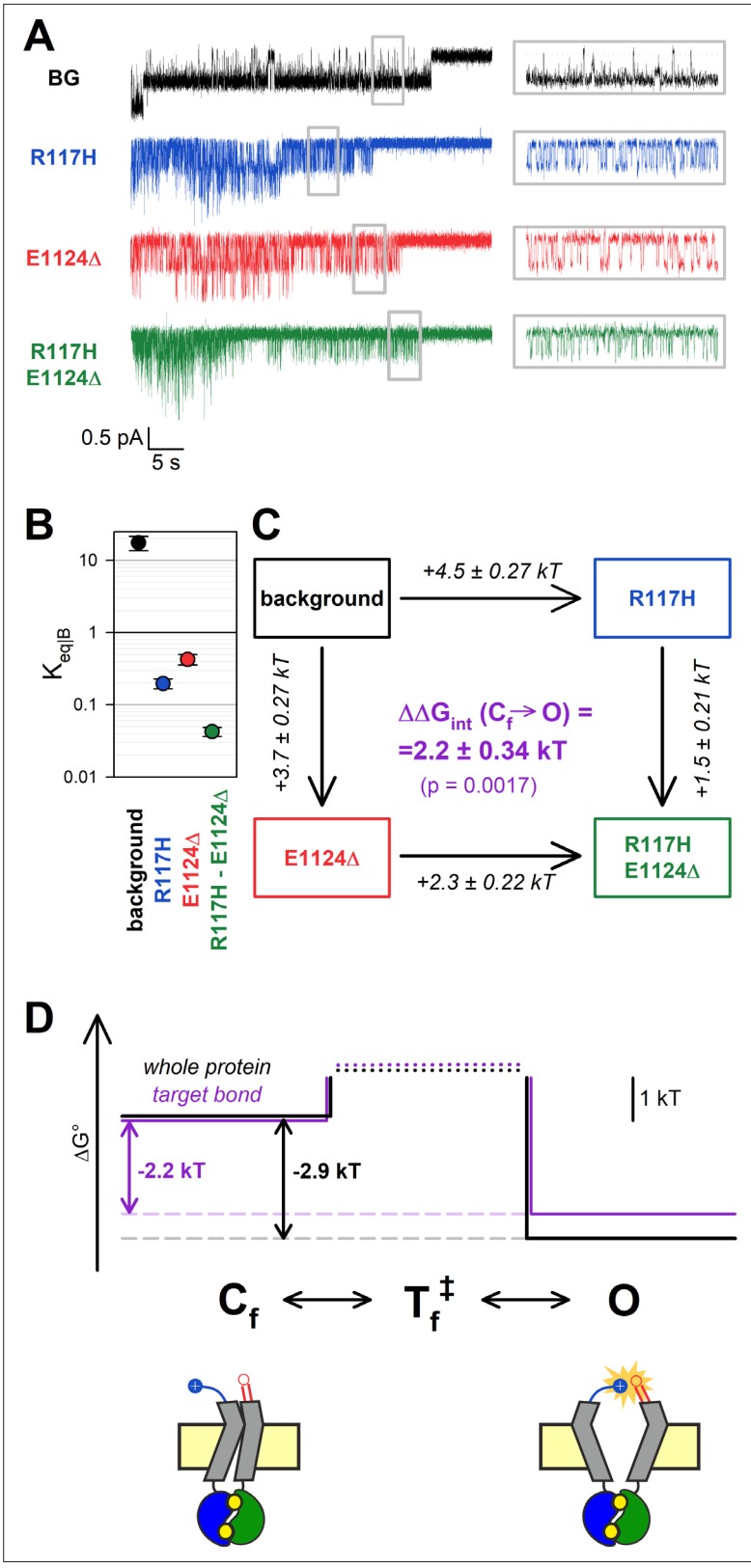

**Figure 4.** The H-bond between the target positions is broken in the flickery closed state. (A) Currents of last open channels surviving ATP removal, for indicated CFTR mutants (*color coded*) harboring the E1371S background mutation. Inside-out patch currents were activated by exposure of pre-phosphorylated channels to 5 mM ATP. Insets (*right*) show 5 s segments corresponding to *gray boxes* at an expanded time scale. $V_m$ was –80 mV. (B)

*Figure 4 continued on next page*

*Figure 4 continued*

Closed-open equilibrium constants within a burst (see Materials and methods). Data are shown as mean ± SEM from five to six patches; note logarithmic ordinate. SEM was calculated by error propagation (see Materials and methods). (C) Thermodynamic mutant cycle showing mutation-induced changes in the stability of the O state relative to the $C_f$ state ($\Delta\Delta G_{O\text{-}Cf}^0$, *numbers on arrows*; $k$, Boltzmann's constant; $T$, absolute temperature). Each corner is represented by the mutations introduced into positions 117 and 1124 of E1371S-CFTR. $\Delta\Delta G_{int}$ (*purple number*) is obtained as the difference between $\Delta\Delta G^0$ values along two parallel sides of the cycle. (D) (*Top*) Free enthalpy profile of the entire channel protein (*black curve*) and of the 117–1124 interaction (*purple curve*) along the states of the intraburst (O↔$C_f$) gating process (see Materials and methods). (*Bottom*) Cartoon representation of channel conformations in the $C_f$ and O states. *Color coding* as in ***Figure 3F***.

within the bursting state, the R117-E1124 H-bond is formed only in the open state, not in the flickery closed state (***Figure 4D***, *purple* bond energy profile; ***Figure 4D bottom***, cartoon).

## Both kinetic schemes used to describe CFTR bursting behavior adequately explain observed effects

CFTR's bursting behavior may be explained by two alternative linear three-state models, $C_{s(low)}\leftrightarrow C_{f(fast)}\leftrightarrow O$ or $C_s\leftrightarrow O\leftrightarrow C_f$. Because those two models cannot be differentiated in any individual steady-state recording, the true model is still unknown. All the analysis presented so far was therefore performed in a model-independent manner. Nevertheless, we also sought to address whether the mutational effects observed here might provide any support for one or the other kinetic mechanism.

Maximum likelihood fits (***Csanády, 2000b***) by the $C_s\leftrightarrow C_f\leftrightarrow O$ and $C_s\leftrightarrow O\leftrightarrow C_f$ model of steady-state single-channel recordings, obtained for the four constructs in the D1370N background (***Figure 3A***), yielded rate constants for all four microscopic transition rates of the respective scheme (***Figure 5C and D***, *center*, average rates (in $s^{-1}$) are *color coded* by mutation). Thus, for both models, a mutant cycle could be built on each of the four transition rates, to quantitate $\Delta\Delta G_{int}$ of the target H-bond for the respective gating step (***Figure 5—figure supplement 1***). Finally, those four $\Delta\Delta G_{int}$ values allowed reconstruction of the entire 117–1124 interaction free enthalpy profile assuming one or the other model (***Figure 5A–B***, *purple plots*). Assuming either model, these profiles predict formation of the target H-bond selectively in state O. Correspondingly, a comparison of the standard free enthalpy profiles of gating for each of the four constructs (***Figure 5C–D***, *top*), calculated for both models from the fitted microscopic transition rate constants (see Materials and methods), unanimously predict a predominant destabilization of the O state by the target-site mutations (***Figure 5C–D***, *top*, colored profiles vs. *black profile*). Moreover, using the $C_s\leftrightarrow O\leftrightarrow C_f$ model the change in target bond strength between states O and $C_s$ (~2.6 kT) and between states O and $C_f$ (~2.5 kT) turned out to be nearly identical (***Figure 5A***). All in all, the data are equally consistent with either model.

## Discussion

Here, we have shown that a strong H-bond between the side chain of R117 and the backbone carbonyl group of E1124 is formed in the open state but neither in the IB nor in the flickery closed state of the CFTR channel. We have further shown that loss of that hydrogen bond is responsible for the strong gating phenotype caused by CF mutation R117H, including a dramatic shortening of burst durations

**Table 2.** Model-independent descriptive parameters of intraburst gating for the indicated mutant CFTR constructs in the E1371S background.

Gating parameters were obtained from last-open-channel recordings as described in Materials and methods. Data are displayed as mean ± SEM from five to six patches.

| | $\tau_{flicker}$ (ms) | $\tau_{open}$ (ms) | $K_{eq|B}$ |
|---|---|---|---|
| Background (E1371S) | 11 ± 2.3 | 200 ± 27 | 17 ± 3.9 |
| R117H | 58 ± 7.3 | 11 ± 1.1 | 0.20 ± 0.031 |
| E1124Δ | 42 ± 6.0 | 18 ± 1.7 | 0.42 ± 0.070 |
| R117H E1124Δ | 150 ± 18 | 6.2 ± 0.50 | 0.042 ± 0.0060 |

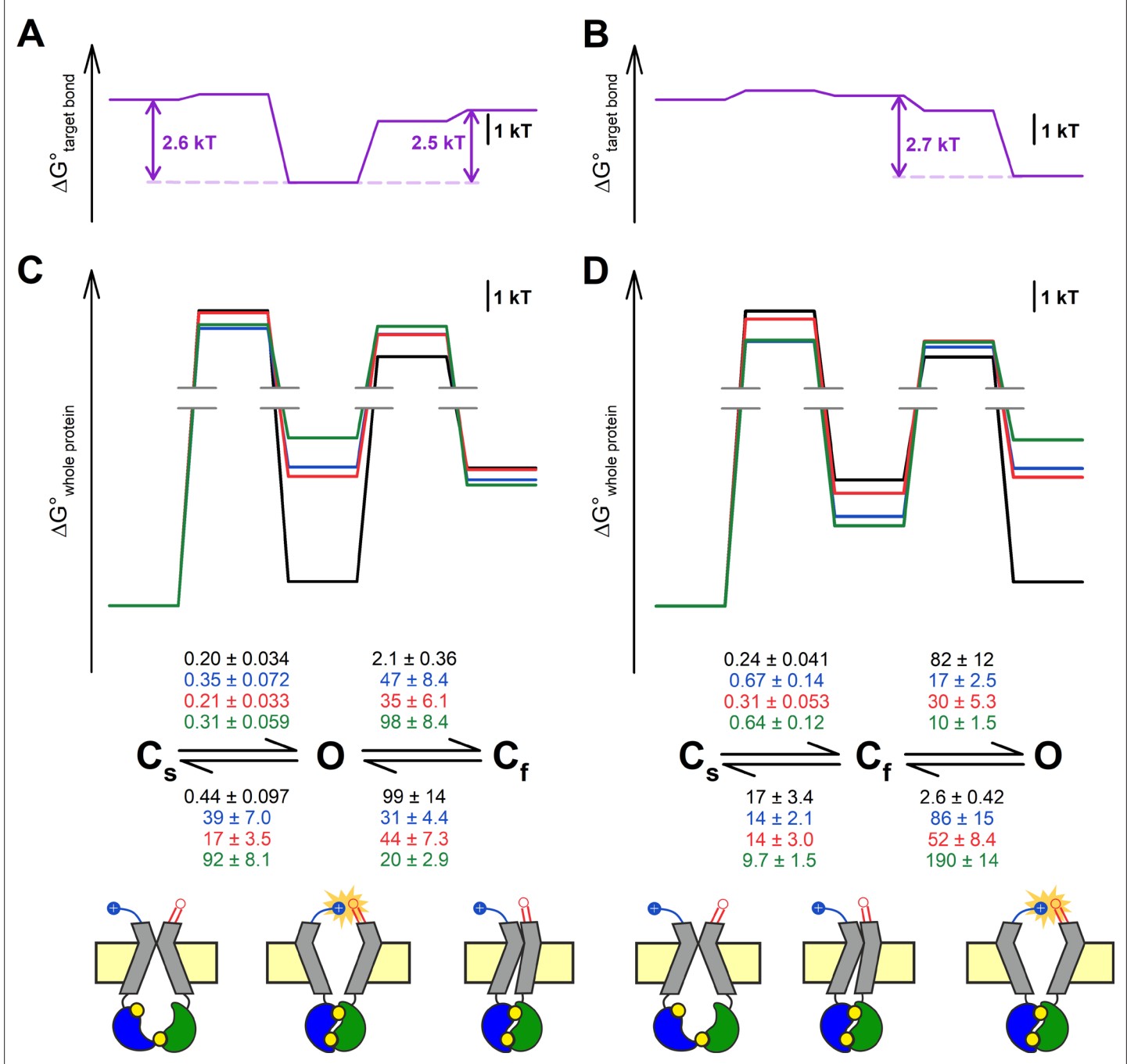

**Figure 5.** Both alternative linear three-state models adequately explain the effects of the mutations. (A–D) Free enthalpy profiles (**A–B**) of the 117–1124 interaction in D1370N-CFTR, and (**C–D**) of the entire channel protein for various mutants (*color coded*) in the D1370N background, along all states of the gating process assuming either the $C_s \leftrightarrow O \leftrightarrow C_f$ (**A, C**) or the $C_s \leftrightarrow C_f \leftrightarrow O$ (**B, D**) model. The profiles in (**A–B**) were obtained from the $\Delta\Delta G_{int}$ values for the four gating steps, those in (**C–D**) from the microscopic transition rate constants (numbers in $s^{-1}$, *color coded* by mutants) obtained by fitting either model to the steady-state recordings in **Figure 3A** (see Materials and methods). Cartoons in C–D (bottom) represent the channel conformations in the $C_s$, O, and $C_f$ states. *Color coding* as in **Figure 3F**.

The online version of this article includes the following figure supplement(s) for figure 5:

**Figure supplement 1.** Mutant cycles built on the four microscopic transition rate constants for the two alternative three-state schemes.

and a strong reduction of intraburst $P_o$. These findings have clarified the molecular pathology of a relatively common CF mutation (and of the less frequent variants R117C/G/L/P) and might prove useful for targeted design of better potentiator drugs for this mutant (see *Ideas and speculation*).

The fact that it is the backbone carbonyl group, not the side chain, of E1124 that participates in this H-bond is consistent with several observations. First, mutations of position 1124 have not been identified in CF patients, consistent with our conclusion that the E1124 side chain does not play an important role. Second, whereas the arginine at the position corresponding to 117 in hsCFTR is absolutely conserved across species, the residue at the position equivalent to 1124 is not (*Figure 2—figure supplement 2*). On the other hand, the *lengths* of both ECL6 and ECL1 are absolutely conserved (*Figure 2—figure supplement 2*), consistent with 3D spacing between those two loops playing an important role.

Based on homology modeling the R117 side chain was earlier proposed to interact with the E1126 side chain (*Cui et al., 2014*). Our data suggest that is not the case, as mutation E1126P which truncates the E1126 side chain did not result in an R117H-like phenotype, but rather prolonged $\tau_{burst}$ (*Figure 2—figure supplement 1*). The reason for the latter effect is unclear, but might be due to changes in flexibility of the ECL6 loop which natively contains three glycines (sequence: GEGEG; *Figure 2—figure supplement 2*). Mutation E1126P (GEG<u>E</u>G → GEG<u>P</u>G) is expected to reduce, whereas mutation E1124G (G<u>E</u>GEG → G<u>G</u>GEG) is expected to increase ECL6 flexibility: correspondingly, $\tau_{burst}$ was prolonged by the former, but slightly shortened by the latter (*Figure 2A–B*, *green*) mutation. Of note, although the cryo-EM structures of quasi-open and closed CFTR do not accurately represent the fully open and the active closed state, respectively, in both structures, the E1126 side chain points away from R117.

Based on the structures the IB→B transition involves a large conformational rearrangement during which dozens of intramolecular interactions are broken and dozens of others are formed (*Liu et al., 2017*; *Zhang et al., 2018a*). How can disruption of a single H-bond result in such a pronounced gating phenotype as that caused by the R117H mutation? Although both conformations of the protein are stabilized by large numbers of local interactions, evolution has carefully balanced the energetic contributions of these, such that net $\Delta G^0$ for the overall IB→B protein conformational change is not far from zero. Indeed, for the D1370N background construct that net $\Delta G^0$ for the entire protein (~1 kT; *Figure 3F*, *black free enthalpy profile*) is smaller than the bond energy of our single target H-bond (~2.6 kT; *Figure 3F*, *purple free enthalpy profile*). Thus, elimination of that single H-bond, selectively present in the B state, can profoundly shift the gating equilibrium toward the IB conformation.

In contrast, based on its fast kinetics, the O↔$C_f$ transition may involve a smaller conformational change with only a few local interactions broken/formed. Because the overall $\Delta G^0$ for that transition is again comparable to that of formation/disruption of the 117–1124 H-bond (~2.9 kT vs. ~2.2 kT in the E1371S background construct; *Figure 4D*, *black* vs. *purple free enthalpy profile*), that bond might represent a key interaction that governs intraburst gating in CFTR. In the cryo-EM structure of phosphorylated ATP-bound zebrafish CFTR (*Zhang et al., 2017*; pdbid: 5w81), although the NBDs are dimerized and the TMDs outward-facing, the external extremity of the pore is tightly sealed. The latter feature is due to a localized displacement of the external portions of TM helices 1 and 12, as compared to the quasi-open structure of human CFTR. Interestingly, that displacement increases the ECL1-ECL6 distance in the zebrafish structure, moving the residues equivalent to our target positions (R118 and D1132) too far away to form an H-bond. Thus, out of two suggested possible alternative explanations (*Zhang et al., 2017*), our findings support the interpretation (*Zhang et al., 2018b*) that the zebrafish outward-facing structure might represent the flickery closed state (cf., cartooned $C_f$ state in *Figures 4–5*).

The magnitude of the effect on B-state stability of disrupting the 117–1124 H-bond depended on the choice of the non-hydrolytic model: $\tau_{burst}$ was shortened by ~6-fold in the E1371S (*Figure 2C–D*), but by ~30-fold in the D1370N (*Figure 3A–B*) background. That discrepancy suggests that the precise arrangement of the ECLs in the B state is not exactly identical in those two constructs, so that the distance between the two target positions, and thus the strength of the H-bond between them, is slightly different (1.4 kT [*Figure 2E*] vs. 2.8 kT [*Figure 3C*]). Which of those two values is more representative of a WT channel? Considering the effect of the R117H mutation on $\tau_{burst}$ of WT CFTR, a semi-quantitative lower estimate can be made. In an earlier study (*Yu et al., 2016*) the rate of the B→IB transition was increased from ~3.5 s$^{-1}$ in WT CFTR to ~11 s$^{-1}$ in R117H CFTR. Since the life time of the posthydrolytic B state is very short (*Csanády et al., 2010*), that rate essentially reflects the life time of the prehydrolytic B state, and is approximated by the sum of the rates for ATP hydrolysis ($k_h$) and for non-hydrolytic closure ($k_{nh}$). In WT channels hydrolytic closure dominates ($k_{nh} << k_h$; *Csanády*

*et al., 2010*), that is, $k_h$ >3 s$^{-1}$, $k_{nh}$ <0.5 s$^{-1}$ (cf., $k_{nh}$~0.5 s$^{-1}$ for the fastest-closing among several known non-hydrolytic models, D1370N [*Figure 3B*, *black symbol*]). Based on the plausible assumption that the ECL1 mutation R117H does not alter the catalytic rate in site 2 of the NBD dimer, in R117H CFTR $k_h$ remains ~3 s$^{-1}$, and thus $k_{nh}$ is ~8 s$^{-1}$. These arguments provide a lowest possible estimate of ~16-fold for the increase in non-hydrolytic closing rate caused by the R117H mutation in WT CFTR, more in line with our findings in the D1370N background.

CFTR bursting may be accounted for by two possible linear three state models, $C_s \leftrightarrow O \leftrightarrow C_f$ and $C_s \leftrightarrow C_f \leftrightarrow O$. In the past, multiple studies have addressed whether ATP dependence (*Winter et al., 1994*), voltage dependence (*Cai et al., 2003*), or pH dependence (*Chen et al., 2017*) of CFTR gating might allow differentiation between those two mechanisms, but in each case both models proved equally adequate to explain the data. Because the 117–1124 H-bond is formed only in state O but not in state $C_s$ or $C_f$ (*Figures 3–4*), we reasoned that mutant cycles built on microscopic transition rates obtained by fits to one or the other model might provide some support for a choice. In particular, if analysis based on the $C_s \leftrightarrow O \leftrightarrow C_f$ model had returned significantly different values of $\Delta\Delta G_{int}$ for the $C_s \leftrightarrow O$ and $C_f \leftrightarrow O$ steps, that would have rendered this model highly unlikely, given that in both of those steps $\Delta\Delta G_{int}$ should reflect the strength of the same H-bond. However, no such discrepancy emerged from our analysis (*Figure 5A*), thus, our data are adequately explained by either model. Moreover, there is currently no structural evidence to support the existence of a fixed order of transitions among the three channel conformations seen in the cryo-EM structures of dephosphorylated human, phosphorylated ATP-bound zebrafish, and phosphorylated ATP-bound human CFTR, which are thought to resemble states $C_s$, $C_f$, and O, respectively (cartooned in *Figure 5C*, bottom). Thus, more structural information will be required to finally settle this question.

## Ideas and speculation

The findings presented here define ECL1-ECL6 as a candidate drug target region on the CFTR protein. Pharmacological stabilization of the open-state interaction between those two loops might efficiently stimulate R117H (and R117C/G/L/P) CFTR channels present in CF patients. Because the target region is extracellular, potentiator compounds acting by such a mechanism would not have to be membrane permeable, that is, could be administered in inhalation form to stimulate CFTR channels in the lung.

## Materials and methods

**Key resources table**

| Reagent type (species) or resource | Designation | Source or reference | Identifiers | Additional information |
|---|---|---|---|---|
| biological sample (*Xenopus laevis*) | *Xenopus laevis* oocytes | European *Xenopus* Resource Centre | RRID: NXR_0.0080 | – |
| Commercial assay or kit | HiSpeed Plasmid Midi Kit | Qiagen | 12643 | |
| Commercial assay or kit | QuickChange II Mutagenesis Kit | Agilent Technologies | 200524 | |
| Commercial assay or kit | mMESSAGE mMACHINE T7 Transcription Kit | ThermoFisher Scientific | AM1344 | |
| Chemical compound, drug | Collagenase type II | ThermoFisher Scientific | 17101–015 | |
| Chemical compound, drug | Adenosine 5'-triphosphoribose magnesium (ATP) | Sigma-Aldrich | A9187 | |
| Chemical compound, drug | Protein kinase A catalytic subunit, bovine | Sigma-Aldrich | P2645 | – |
| Chemical compound, drug | Vx-770 (solid) | Selleck Chemicals | S1144 | – |
| Chemical compound, drug | $N^6$-(2-phenylethyl)-dATP | Biolog LSI | D 104 | – |
| Software, algorithm | Pclamp9 | Molecular Devices | RRID: SCR_011323 | |

## Molecular biology

Mutations were introduced into the human CFTR(E1371S)/pGEMHE and CFTR(D1370N)/pGEMHE sequences using the QuikChange II XL Kit (Agilent Technologies) and confirmed by automated sequencing (LGC Genomics GmbH). To generate cRNA, the cDNA was linearized (Nhe I HF, New England Biolabs) and transcribed in vitro using T7 polymerase (mMessage mMachine T7 Kit, Thermo Fisher), cRNA was stored at –80°C.

## Functional expression of human CFTR constructs in *Xenopus laevis* oocytes

Oocytes were removed from anesthetized *Xenopus laevis* following Institutional Animal Care Committee guidelines, and separated by collagenase treatment (Collagenase type II, Gibco). Isolated oocytes were kept at 18°C, in a modified frog Ringer's solution (in mM: 82 NaCl, 2 KCl, 1 MgCl$_2$, and 5 HEPES, pH 7.5 with NaOH) supplemented with 1.8 mM CaCl$_2$ and 50 µg/ml gentamycin. Injections by 0.1–10 ng of cRNA, to obtain microscopic or macroscopic currents, was done in a fixed 50 nl volume (Nanoject II, Drummond). Recordings were performed 1–3 days after injection.

## Excised inside-out patch recording

The patch pipette solution contained (in mM): 138 NMDG, 2 MgCl$_2$, 5 HEPES, pH = 7.4 with HCl. The bath solution contained (in mM): 138 NMDG, 2 MgCl$_2$, 5 HEPES, 0.5 EGTA, pH = 7.1 with HCl. Following excision into the inside-out configuration patches were moved into a flow chamber in which the composition of the continuously flowing bath solution could be exchanged with a time constant of <100 ms using electronic valves (ALA-VM8, Ala Scientific Instruments). CFTR channel gating was studied at 25°C, in the presence of 5 mM MgATP (Sigma), following activation by ~1 min exposure to 300 nM bovine PKA catalytic subunit (Sigma P2645). Macroscopic currents were recorded at –20 mV, microscopic currents at –80 mV membrane potential. For the low-P$_o$ mutants R117H, E1124Δ, and R117H-E1124Δ, evaluation of the number of active channels in the patch was facilitated by stimulating the channels with 50 µM $N^6$-(2-phenylethyl)-dATP (P-dATP; Biolog LSI), with or without 10 nM Vx-770 (Selleck Chemicals), at the end of each experiment. Currents were amplified and low-pass filtered at 1 kHz (Axopatch 200B, Molecular Devices), digitized at a sampling rate of 10 kHz (Digidata 1550B, Molecular Devices) and recorded to disk (Pclamp 11, Molecular Devices).

## Kinetic analysis of electrophysiological data

To obtain relaxation time constants (*Figure 2B and D*), macroscopic current relaxations were fitted to single exponentials by non-linear least squares (Clampfit 11).

For kinetic analysis of steady-state single-channel recordings in the D1370N background, baseline-subtracted unitary currents, from recordings with no superimposed channel openings, were Gaussian-filtered at 100 Hz and idealized by half-amplitude threshold crossing. Recordings were further analyzed only if the presence of a second active channel could be excluded with >90% confidence using statistical tests described earlier (*Csanády et al., 2000a*). All events lists were fitted with both the $C_{1(s)} \leftrightarrow O_3 \leftrightarrow C_{2(f)}$ and the $C_{1(s)} \leftrightarrow C_{2(f)} \leftrightarrow O_3$ model using a maximum likelihood approach which accounted for the presence of an imposed fixed dead time of 4 ms, and returned a set of microscopic transition rate constants $k_{12}$, $k_{21}$, $k_{23}$, and $k_{32}$ for both models (*Csanády, 2000b*). As verified on simulated data sets analyzed at the same bandwidth, these rate estimates remained unbiased even for the low-P$_o$ constructs. The descriptive kinetic parameters $\tau_{burst}$, $\tau_{interburst}$, $\tau_{flicker}$, and $n_{flicker}$ (average number of flickery closures per burst) were calculated from the rate constants as follows *Colquhoun and Sigworth, 1995*. For the $C_1 \leftrightarrow O_3 \leftrightarrow C_2$ model $\tau_{burst}=(1/k_{31})(1+k_{32}/k_{23})$, $\tau_{interburst}=1/k_{13}$, $\tau_{flicker}=1/k_{23}$, $n_{flicker}=k_{32}/k_{31}$. For the $C_1 \leftrightarrow C_2 \leftrightarrow O_3$ model $\tau_{burst}=(1/k_{21})((k_{21}+k_{23})/k_{32}+k_{23}/(k_{21}+k_{23}))$, $\tau_{interburst}=((k_{12}+k_{21}+k_{23})/(k_{12}k_{23})) + 1/(k_{21}+k_{23})$, $\tau_{flicker}=1/(k_{21}+k_{23})$, $n_{flicker}=k_{23}/k_{21}$. Importantly, fits using either model necessarily yield identical sets of calculated parameters $\tau_{burst}$, $\tau_{interburst}$, $\tau_{flicker}$, and $n_{flicker}$, thus, the latter descriptive parameters (*Figure 3B–D*, *Table 1*) are model-independent. The rate of the IB→B step was taken as $1/\tau_{interburst}$, and that of the B→IB step as $1/\tau_{burst}$.

For intraburst kinetic analysis of the last open channel in the E1371S background, mean open times ($\tau_{open}$) and mean flickery closed times ($\tau_{flicker}$) were obtained as the simple arithmetic averages of the mean open and closed dwell-time durations, respectively, and $K_{eq|B}$ was calculated as $K_{eq|B}=\tau_{open}/\tau_{flicker}$ (*Figure 4B*, *Table 2*).

## Mutant cycle Analysis

Changes in the strength of the R117-E1124 interaction between various gating states were evaluated using thermodynamic mutant cycle analysis as described (*Mihályi et al., 2016*). For an $S_1 \leftrightarrow T^\ddagger \leftrightarrow S_2$ gating step mutation-induced changes in the relative stabilities of those three states were calculated as follows. The change in the height of a transition-state barrier, for example, for step $S_1 \rightarrow T^\ddagger$, was calculated as $\Delta\Delta G^0_{T^{++}-S_1} = -kTln\left(r'_{12}/r_{12}\right)$, where $k$ is Boltzmann's constant, $T$ is absolute temperature, and $r_{12}$ and $r'_{12}$ are the rates for the $S_1 \rightarrow S_2$ transition in the background construct and in the mutant, respectively. The change in the stability of the two ground states relative to each other was calculated as $\Delta\Delta G^0_{S_2-S_1} = -kTln\left(K'_{eq}/K_{eq}\right)$, where $K_{eq}$ and $K'_{eq}$ are the equilibrium constants for the $S_1 \leftrightarrow S_2$ transition in the background construct and in the mutant, respectively. Interaction free energy ($\Delta\Delta G_{int}$) was defined as the difference between $\Delta\Delta G^0$ values along two parallel sides of a mutant cycle. All $\Delta\Delta G$ values are given as mean ± SEM; SEM values were estimated assuming that $r_{ij}$ and $K_{eq}$ are normally distributed random variables, using second-order approximations of the exact integrals (*Mihályi et al., 2016*).

## Calculation of free enthalpy profiles

To construct standard free enthalpy profiles for channel gating (*Figures 3F and 4D*, *black profiles*; *Figure 5C–D*) $\Delta\Delta G^0$ between two ground states was calculated as $\Delta\Delta G^0 = -kT \ln K_{eq}$. For example, for *Figure 3F* $\Delta\Delta G^0(IB \rightarrow B) = -kT \ln(\tau_{burst}/(\tau_{interburst}))$, values for $\tau_{burst}$ and $\tau_{interburst}$ are given in *Table 1*; for *Figure 4D*, $\Delta\Delta G^0(C_f \rightarrow O) = -kT \ln(\tau_{open}/(\tau_{flicker}))$; values for $\tau_{open}$ and $\tau_{flicker}$ are given in *Table 2*. The SEM for $K_{eq|B}$ in *Figure 4B* was calculated by error propagation from the variances of $\tau_{open}$ and $\tau_{flicker}$. The absolute values of the barrier heights were not calculated (see breaks), but the relative heights of the barriers were appropriately scaled as follows. The difference between the heights of two alternative barriers for exiting a single source state was calculated as $\Delta\Delta G^0_{T_1^{++}-T_2^{++}} = -kTln\left(r_1/r_2\right)$, where $r_1$ and $r_2$ are the rates of exit along those two pathways. Similarly, the difference between the heights of a given barrier for two different constructs was calculated using the same equation, but with $r_1$ and $r_2$ reflecting the corresponding transition rates for constructs 1 and 2, respectively.

## Statistics

All values are given as mean ± SEM, with the numbers of independent samples provided in each figure legend. Statistical significances of interaction free energies were calculated using Student's t test, $\Delta\Delta G_{int}$ is reported significantly different from zero for $p < 0.05$.

## Acknowledgements

Supported by EU Horizon 2020 Research and Innovation Program grant 739593, MTA Lendület grant LP2017-14/2017, and Cystic Fibrosis Foundation Research Grant CSANAD21G0 to LC. MAS received support from the ÚNKP-20–3-I-SE-34 New National Excellence Program of the Ministry for Innovation and Technology from the source of the National Research, Development and Innovation Fund.

# Additional information

## Funding

| Funder | Grant reference number | Author |
|---|---|---|
| Magyar Tudományos Akadémia | LP2017-14/2017 | László Csanády |
| Cystic Fibrosis Foundation | CSANAD21G0 | László Csanády |
| Ministry for Innovation and Technology | ÚNKP-20-3-I-SE-34 | Márton A Simon |
| European Union | 739593 | László Csanády |

The funders had no role in study design, data collection and interpretation, or the decision to submit the work for publication.

### Author contributions
Márton A Simon, Formal analysis, Funding acquisition, Investigation, Software, Visualization, Writing – original draft; László Csanády, Conceptualization, Data curation, Funding acquisition, Methodology, Project administration, Software, Supervision, Validation, Writing – original draft

### Author ORCIDs
Márton A Simon (ID) http://orcid.org/0000-0002-5411-9408
László Csanády (ID) http://orcid.org/0000-0002-6547-5889

### Ethics
This study was performed in strict accordance with the recommendations in the Guide for the Care and Use of Laboratory Animals of the National Institutes of Health. All of the animals were handled according to approved institutional animal care and use committee (IACUC) protocols of Semmelweis University (last approved 06-03-2021, expiration 06-03-2026).

### Decision letter and Author response
Decision letter https://doi.org/10.7554/eLife.74693.sa1
Author response https://doi.org/10.7554/eLife.74693.sa2

## Additional files

### Supplementary files
• Transparent reporting form

### Data availability
All data generated or analysed during this study are included in the main text figures, tables, and supporting figures.

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
