## [Editor Report]

Multiple inherited mutations in the epithelial CFTR anion-permeable channel cause cystic fibrosis through different molecular mechanisms, and some of these mechanisms can be specifically targeted by drugs to treat the disease. Drawing from available structural information and double-mutant cycle analysis applied to patch-clamp recordings, Simon and Csanády find that one of the most common CFTR disease-causing mutations, R117H, disrupts an interaction between the R117 side-chain and a main-chain carbonyl that selectively stabilizes the open state of the channel. These findings may open new paths of exploration for treating patients carrying this mutation, and provide important mechanistic constraints towards understanding the gating mechanism of CFTR channels.

---

## [Decision Letter]

**Decision letter after peer review:**

Thank you for submitting your article "Molecular pathology of a cystic fibrosis mutation is explained by loss of a hydrogen bond" for consideration by *eLife*. Your article has been reviewed by 3 peer reviewers, including Andrés Jara-Oseguera as the Reviewing Editor and Reviewer #1, and the evaluation has been overseen by Richard Aldrich as the Senior Editor. The following individual involved in review of your submission has agreed to reveal their identity: Tzyh-Chang Hwang (Reviewer #2).

The reviewers have discussed their reviews with one another, and are all in agreement that the study is interesting and that the data and analysis are of high quality. However, the reviewers also identified a series of weaknesses in the study that would need to be addressed, the main one being the lack of direct structural evidence in support of the open-state dependence of the interaction between R117 and the main-chain carbonyl of residue E1224. Based on the discussion between reviewers, the Reviewing Editor has drafted this to help you prepare a revised submission.

Essential revisions:

Title:

1. The title should be improved. As the authors stated in the abstract and introduction, CFTR has a wide variety of mutations and a broad range of pathologies. This paper focuses on the R117-E1124 interaction and therefore does not reflect all molecular pathologies of CFTR. A title that reflects the focus of this paper is considered more appropriate.

Introduction:

1. P3 first sentence: Gut and intestine are the same thing.

2. P3 second sentence: The actual CF-causing mutations are less than 1700. Many of the mutations in the data bank are polymorphisms.

3. P3: The sentence "Treatment by.…" may need revision.

4. P3 second sentence of the second paragraph: Please spell out the R domain.

5. P3: I would caution the authors not to use "inward-facing" and "outward-facing" to describe CFTR gating conformational changes since there is really no typical outward-facing configuration in CFTR.

6. P4: The major reason for a drastic reduction of the Po by the R117H mutation is actually not a shortening of the burst duration, but a > 3-fold prolongation of the interburst duration (see Yu et al., 2016). The authors seem to discount this observation throughout the manuscript. It is therefore not appropriate to claim that "loss of this hydrogen bond explains the entire gating phenotype of the R117H mutant" in the Abstract. This should be addressed in the manuscript text.

Results:

1. P6: Please specify the duration of exposure to 5 mM ATP.

2. P7, line 8: Without extensive studies of E1124A, it is probably premature to claim "the E1124 side chain is not required for normal gating."

3. P8 last sentence: The D1370N mutation does not "remove" the Walker B glutamate side chain.

4. P9: It seems that the τ_interburst_ for D1370N in the current study is much larger than one reported previously (5 s versus 2 s in Sorum et al., 2017). Previous work was done in a background that is independent of phosphorylation. Doesn't this discrepancy implicate the importance of phosphorylation on the estimated τ_interburst_? Indeed, the model proposed in Mihalyi et al. (2020) also suggests that the level of phosphorylation in the R domain can affect τ_interburst_. In light of a lack of stringent control of phosphorylation in the oocyte expression system (Csanady et al., 2013) and the effect of R domain phosphorylation on the tinterburst, some discussion over this caveat is needed. This, plus the fact that extreme long interburst times pose a severe limit on the number of events collected (Figure 3-Figure Suppl 2), makes it very difficult to accurately estimate the τ_interburst_. This uncertainty out of technical and conceptual considerations should be borne in mind while discussing the opening rate.

5. Figure 3F: Please explain how 0.96 kT was obtained.

6. P11, line 8-10: Please revise.

7. It would strengthen the conclusions by the authors if they included in Figure 1B the experimental densities associated with the structures, to provide further support for the interaction on which the manuscript is based.

8. The authors should provide more information regarding the computational work. From the main text, it appears that the authors carried out molecular dynamics simulations, but the information in the methods suggests that the authors carried out molecular modeling. If MD simulations were carried out, it would strengthen the manuscript if the authors provided additional data with the MD trajectories showing that the structures on which the calculations were carried out are stable, and that the minimal distance shown in Figure 2 Suppl. 2 has the highest occupancy. If multiple models were compared, it would be also beneficial if the authors provided a distribution of the calculated distances in the top-scoring models, rather than simply showing the mean value.

9. The authors should comment on the potential uncertainty in estimating the intraburst gating parameters from the data in Figure 3A for the constructs with low open probability.

10. The authors should comment on the effects of R117H or E1124Δ on the intraburst kinetic parameters (Figure 4), which appear to suggest that the introduced structural perturbations are not entirely limited to the open state of the channel.

11. For all group data presented in figures (especially bar graphs), the authors should not only show the mean+/-S.E.M but also the individual data points.

Discussion:

1. It is clear that recent structure biological studies of CFTR have provided exquisite mechanistic insights into CFTR function as a phosphorylation-activated, ATP-gated chloride channel. However, the structures solved so far also raise a plethora of questions. Moreover, equating the unphosphorylated closed conformation to phosphorylated closed state seems problematic as previous studies suggest that the two NBDs in phosphorylated CFTR may not assume a conformation that is widely separated. While I applaud the authors' attempt to connect the dots between CFTR functional states and the available structures, I also believe that a more conservative approach is more appropriate at this stage. I therefore suggest that authors tone down their speculations in the Discussion.

2. P13, lines 353-356: A two-armed scale and a protein are energetically different systems. The authors are clearly stating their points in the paragraph already even without the metaphor. The metaphor is not necessary.

3. Since authors are mentioning the implications of the study for CF treatment with targeted potentiator drugs in the Abstract and Introduction, it would be interesting for readers if authors included their opinion on how the findings presented here could help to develop a potentiator drug. *eLife* now offers authors the opportunity to include an "Ideas and Speculation" subsection within their Discussion, which would enable the authors to offer some amount of speculation in this regard.

---

## [Author Response]

Essential revisions:Title:1. The title should be improved. As the authors stated in the abstract and introduction, CFTR has a wide variety of mutations and a broad range of pathologies. This paper focuses on the R117-E1124 interaction and therefore does not reflect all molecular pathologies of CFTR. A title that reflects the focus of this paper is considered more appropriate.

We have made the title more focused by specifying the CF mutation in focus:

"Molecular pathology of the R117H cystic fibrosis mutation is explained by loss of a hydrogen bond".

Introduction:1. P3 first sentence: Gut and intestine are the same thing.

Amended, thank you.

2. P3 second sentence: The actual CF-causing mutations are less than 1700. Many of the mutations in the data bank are polymorphisms.

Thank you, corrected for "several hundred".

3. P3: The sentence "Treatment by.…" may need revision.

Revised, thank you.

4. P3 second sentence of the second paragraph: Please spell out the R domain.

Done, thank you.

5. P3: I would caution the authors not to use "inward-facing" and "outward-facing" to describe CFTR gating conformational changes since there is really no typical outward-facing configuration in CFTR.

We agree, and have actually discussed this issue in detail (original lines 97-100). Nevertheless, the structures do fall into two distinct groups: structures with separated NBDs and an inward-facing arrangement of TMDs, and structures with dimerized NBDs and an overall outward-facing arrangement of the TMDs. We now provide a more in-depth coverage of the limitations associated with the structures, and provide a clear definition for the nomenclature adopted in this paper:

"… the former structure (6msm) will be referred to as "outward-facing" or "quasi-open", and the latter (5uak) as "inward-facing" or "closed" throughout this study". (lines 103-115).

6. P4: The major reason for a drastic reduction of the Po by the R117H mutation is actually not a shortening of the burst duration, but a > 3-fold prolongation of the interburst duration (see Yu et al., 2016). The authors seem to discount this observation throughout the manuscript. It is therefore not appropriate to claim that "loss of this hydrogen bond explains the entire gating phenotype of the R117H mutant" in the Abstract. This should be addressed in the manuscript text.

In the Yu et al., 2016 study a direct comparison of WT and R117H opening rates (Figure 1B) revealed a ~25% decrease in opening rate for R117H relative to WT (interburst duration was ~300 ms for WT but ~400 ms for R117H), but the difference was claimed insignificant ("the R117H mutation probably also decreases the opening rate, although the difference in the closed time constant between WT- and R117H-CFTR is not yet statistically significant (Figure 1B)"). In contrast, in the same figure the mean burst duration of R117H was shown to be ~3-fold shorter than that of WT CFTR (~300 ms for WT but ~100 ms for R117H). These findings are in good agreement with those described in our study.

Although in later figures of the Yu et al. 2016 study the authors obtained somewhat smaller opening rates for R117H when accurate channel counting was facilitated by stimulating the channels with Vx770, no direct comparison was performed to evaluate how estimation of opening rate for WT CFTR is affected when a similar approach for channel counting is employed. Therefore, although the data in Yu et al. suggested that opening rate might perhaps be slightly affected by the R117H mutation, no direct proof for that hypothesis was provided. The Reviewer's claim that the "major reason for a drastic reduction of the Po by the R117H mutation.. is… a prolongation of the interburst duration" is therefore inconsistent both with the data in the Yu et al. 2016 study and those presented here. (Moreover, the ~3fold reduction in τ_b_ together with the drastic reduction in intraburst P_o_ reported by Yu et al. adequately accounts for the ~6-fold reduction of CFTR whole-cell conductance by the R117H mutation observed earlier (Sheppard et al., 1993), without the postulation of a slowed opening rate.)

Of note, in the present study, and also in Sorum et al. 2017, estimation of opening rates was based exclusively on patches in which channel number could be estimated with high confidence following strong stimulation by Vx-770 and/or P-dATP (Figure 3—figure supplement 3) – these studies did not reveal any significant reduction of opening rate by the R117H mutation.

Nevertheless, we have changed the word "*entire*" to "*strong*" in the Abstract, and also acknowledge limitations of accurately estimating channel opening rate (lines 270-272).

Results:1. P6: Please specify the duration of exposure to 5 mM ATP.

The duration of exposure to 5 mM ATP was ~ 1 minute, this has been added to the text (line 180).

2. P7, line 8: Without extensive studies of E1124A, it is probably premature to claim "the E1124 side chain is not required for normal gating."

We agree. We have toned down that claim:

"the E1124 side chain is not required for a normal burst duration" (line 193).

3. P8 last sentence: The D1370N mutation does not "remove" the Walker B glutamate side chain.

Thank you, moreover, D1370 is an aspartate not a glutamate. We now say "which perturbs the Walker B aspartate side chain" (line 242).

4. P9: It seems that the τ_interburst_ for D1370N in the current study is much larger than one reported previously (5 s versus 2 s in Sorum et al., 2017). Previous work was done in a background that is independent of phosphorylation. Doesn't this discrepancy implicate the importance of phosphorylation on the estimated τ_interburst_? Indeed, the model proposed in Mihalyi et al. (2020) also suggests that the level of phosphorylation in the R domain can affect τ_interburst_. In light of a lack of stringent control of phosphorylation in the oocyte expression system (Csanady et al., 2013) and the effect of R domain phosphorylation on the tinterburst, some discussion over this caveat is needed.

The τ_ib_ values reported for the D1370N mutant in various studies are indeed somewhat variable. The estimate in the present study is ~2x longer than that reported in Sorum et al. (2017) or Csanády et al. (2010), but comparable to that reported in Vergani et al. (2003) (all of those studies were done in the *Xenopus* oocyte expression system). Although phosphorylation does affect opening rate, we do not think that the above scatter is attributable to differential degrees of phosphorylation. The "lack of stringent control of phosphorylation in the oocyte expression system (Csanady et al., 2013)" mentioned by the Reviewer refers to the rapid partial current decline which is observed immediately upon PKA removal in inside-out patches excised from *Xenopus oocytes* and was earlier interpreted to reflect rapid dephosphorylation of a subset of phosphoserines in the R domain. However, the recent study by Mihalyi et al. (2020) has demonstrated that the rapid partial current decline reflects immediate loss of a direct stimulation provided by PKA binding, whereas endogenous phosphatase activity in the patches is minimal. Thus, in retrospect, in all of the above studies D1370N channels were likely fully phosphorylated. The differences are more likely to reflect biological variability due to unknown factors. Of note, such degree of variability is typical for biological systems, and is not inherent to the *Xenopus* oocyte expression system or the D1370N construct. (E.g., τ_ib_ values ranging from 300 ms (Yu et al., 2016) to 500 ms (Tsai et al., 2010) have been reported for WT CFTR expressed in CHO cells.)

This, plus the fact that extreme long interburst times pose a severe limit on the number of events collected (Figure 3-Figure Suppl 2), makes it very difficult to accurately estimate the τ_interburst_. This uncertainty out of technical and conceptual considerations should be borne in mind while discussing the opening rate.

In each individual patch the numbers of interburst events analyzed were sufficient to estimate τ_ib_ to within <20% of the mean value (for an exponential distribution S.E.M. = mean/sqrt(n); i.e., already for 25 events S.E.M. = 0.2*mean), and 5-8 patches were analyzed for each construct. Nevertheless, we agree that estimation of opening rates is more challenging than estimation of closing rates due to the uncertainty in the number of active channels, which can never be completely eliminated. This caveat was already discussed in the Results section (original lines 263-268), but we have now added the following sentence:

"Nevertheless, compared to τ_burst_, estimates of τ_interburst_ necessarily retain some degree of uncertainty, as the number of active channels can never be determined with 100% confidence (cf., (Yu et al., 2016))" (lines 270-272).

5. Figure 3F: Please explain how 0.96 kT was obtained.

This is described in Materials and methods (line 497, now expanded on lines 497-500):

∆∆G^0^(IB→B) = -*kT* ln(*K*_eq_) = -*kT* ln(*P*_burst_/(1-*P*_burst_)) = -*kT* ln(τ_burst_/(τ_interburst_)) Values for τ_burst_ and τ_interburst_ are given in Table 1.

6. P11, line 8 -10: Please revise.

Based on the explanation provided in our response to comment (10.), below, we believe that this sentence is adequately worded as written.

7. It would strengthen the conclusions by the authors if they included in Figure 1B the experimental densities associated with the structures, to provide further support for the interaction on which the manuscript is based.

Done. A close-up view of the R117-E1124 interaction in the quasi-open human CFTR structure is shown with the electron density in Fig. 1—figure supplement 1. Both loops are well represented, and density for the R117 side chain is visible down to the delta carbon.

8. The authors should provide more information regarding the computational work. From the main text, it appears that the authors carried out molecular dynamics simulations, but the information in the methods suggests that the authors carried out molecular modeling. If MD simulations were carried out, it would strengthen the manuscript if the authors provided additional data with the MD trajectories showing that the structures on which the calculations were carried out are stable, and that the minimal distance shown in Figure 2 Suppl. 2 has the highest occupancy. If multiple models were compared, it would be also beneficial if the authors provided a distribution of the calculated distances in the top-scoring models, rather than simply showing the mean value.

We acknowledge that the way we used the term "molecular dynamics simulations" was misleading. As described in Methods, our approach should be better referred to as modeling, although the program Modeler implements a molecular dynamics simulation run in the final optimization of loops. Although the results of these calculations rationalized our experimental approach, we have decided to omit this supplementary figure for two reasons. First, the strategy of using a deletion to try to shorten a loop is a trivial idea which does not require in silico modeling. Second, our use of Modeler for this purpose is unconventional, and would require extensive validation and testing to establish its reliability.

9. The authors should comment on the potential uncertainty in estimating the intraburst gating parameters from the data in Figure 3A for the constructs with low open probability.

The dead time of our low-pass (Gaussian) filter was 1.8 ms, but a 4-ms artifical dead time was imposed on the events lists to eliminate dwell-times with durations between t_d_ and ~2t_d_ that are distorted by filtering (Colquhoun and Sigworth, 1995). This bandwidth limitation indeed truncates the dwell-time distributions of the short-burst constructs (see Figure 3—figure supplement 2.). However, the ML fitting procedure employed for the extraction of gating parameters from the dwell-time distributions includes a robust correction for missed events, which was extensively verified on simulated data (Csanády, 2000). Thus, the obtained steady-state kinetic parameters (Figure 5) are reasonably reliable and allowed us to build mutant cycles even on the intraburst gating parameters (Figure 5—figure supplement 1). Of note, whereas interpretation of IB↔B gating rates (Figure 3) is model-independent, interpretation of the intraburst gating rates is model-dependent – as evident from Figures 5C-D. We have included a comment on the reliability of our kinetic analysis in methods (lines 463-465).

(In contrast, the dwell-time analysis of last open channels in the E1371S background (Figure 4A-B and Table 2) is based on simple mean intraburst open and closed times, which might be indeed slightly distorted by the 1.8-ms filter dead time. That is why we refrained from analyzing the intraburst gating rates in Figure 4, and provide only an analysis off K_eq|B_ values which are (i) less sensitive to filtering, and (ii) also model-independent.)

10. The authors should comment on the effects of R117H or E1124Δ on the intraburst kinetic parameters (Figure 4), which appear to suggest that the introduced structural perturbations are not entirely limited to the open state of the channel.

The Reviewers point out that "if the interaction occurs only in the open state, then the transition from the C_f_ state to the open state should not be affected by any of the perturbations, but this rate seems to also become altered". The Reviewers are correct that the mutations affect not only the mean open time (τ_open_) but also the mean flickery closed time duration (τ_flicker_) (Table 2); a similar pattern is observed also in the D1370N background (Table 1). However, the inference of an underlying stabilization of the flickery closed state by the H-bond is incorrect. Two points must be considered here.

i) The rate of opening from the flickery closed state reflects the height of the energy barrier (∆∆G_Tf-Cf_). The mutation-induced prolongations of τ_flicker_ suggest that the mutations destabilize not only ground state O, but to some extent also the transition state T_f_^‡^, relative to ground state C_f_.

ii) Prolongation of τ_flicker_ by a mutation at either position 117 or 1124 provides little information regarding the role of the R117-E1124 H-bond, because in the WT protein the targeted residue might be involved in multiple interactions, all of which are altered by its mutation. To dissect and quantify the effect on τ_flicker_ of perturbing the target H-bond, thermodynamic mutant cycle analysis must be employed (see Page 7, bottom). From our analysis the energetic effects of the R117H and E1124∆ mutations on rate C_f_→O appeared largely additive in the double mutant, i.e., ∆∆G_int_(C_f_→T_f_^‡^) was not significantly different from zero, regardless of the chosen gating model (Figure 5—figure supplement 1D, G). This suggests that the H-bond is formed neither in state C_f_ nor in state T_f_^‡^, but exclusively in state O, as depicted in the free energy profiles in Figure 5A-B.

11. For all group data presented in figures (especially bar graphs), the authors should not only show the mean+/-S.E.M but also the individual data points.

Done. (Except for Figure 4B, in which S.E.M. values were calculated by error propagation; see Figure 4 legend and Materials and methods.)

Discussion:1. It is clear that recent structure biological studies of CFTR have provided exquisite mechanistic insights into CFTR function as a phosphorylation-activated, ATP-gated chloride channel. However, the structures solved so far also raise a plethora of questions. Moreover, equating the unphosphorylated closed conformation to phosphorylated closed state seems problematic as previous studies suggest that the two NBDs in phosphorylated CFTR may not assume a conformation that is widely separated.

Both the discrepancy between the outward-facing CFTR structure and the CFTR open state and that between the inward-facing structure and the CFTR closed state pointed out by the Reviewer have been discussed in our manuscript (original lines 97-100 and 155-158). In fact, it is exactly these shortcomings of the available structures that lend significance to our functional studies. Thus, it was not our intention to "equate" CFTR gating states with the existing structures, but rather to perform functional studies to test the validity and functional relevance of a putative interaction suggested by those structures. "The latter structure does not represent the closed state of an active channel gating in the presence of ATP… the identified 117-1124 H-bond might represent an interaction that changes dynamically… To study the functional relevance of this putative interaction…" (original lines 155-162). We acknowledge that all this information appeared somewhat scattered in the original text, which must have been confusing to the reader. To clarify these issues we have moved this entire information to the Introduction, and now devote a separate paragraph to the exposure of limitations associated with the available structures, as well as the nomenclature adopted in our study (lines 103115):

"The cryo-EM studies have provided unprecedented structural insight, but have left several questions open. […] Thus, the former structure (6msm) will be referred to as "outward-facing" or "quasi-open", and the latter (5uak) as "inward-facing" or "closed" throughout this study."

While I applaud the authors' attempt to connect the dots between CFTR functional states and the available structures, I also believe that a more conservative approach is more appropriate at this stage. I therefore suggest that authors tone down their speculations in the Discussion.

Discussion is focused on the functional aspects of our findings, cryo-EM structures are discussed in only two instances. One of these regarded the positioning of side chain E1126 in the open and closed state. The reference to the cryo-EM structures is actually not needed here, as our functional data suffice to rule out the involvement of the E1126 side chain in forming an H-bond with R117. The structures are now mentioned on lines 349-351, in a cautiously reworded note. The other instance in which a cryo-EM structure is referred to, that of zebrafish CFTR (lines 366-375), is adequately worded, as we do not make any firm statement here. We simply note that our functional data support one of two earlier interpretations of the structure. We believe that this level of "speculation" is adequate for a Discussion section.

2. P13, lines 353-356: A two-armed scale and a protein are energetically different systems. The authors are clearly stating their points in the paragraph already even without the metaphor. The metaphor is not necessary.

We have deleted the metaphor.

3. Since authors are mentioning the implications of the study for CF treatment with targeted potentiator drugs in the Abstract and Introduction, it would be interesting for readers if authors included their opinion on how the findings presented here could help to develop a potentiator drug. eLife now offers authors the opportunity to include an "Ideas and Speculation" subsection within their Discussion, which would enable the authors to offer some amount of speculation in this regard.

Done. ("Ideas and Speculation", lines 411-417.)